# Internists' ambivalence toward their role in health counseling and promotion: A qualitative study on the internists' beliefs and attitudes

Nicolien M. H. Kromme[1]*, Kees T. B. Ahaus[2], Reinold O. B. Gans[1], Harry B. M. van de Wiel[3]

1 Department of Internal Medicine, University Medical Center Groningen, University of Groningen, Groningen, The Netherlands, 2 Erasmus School of Health Policy & Management Health Services Management & Organization (HSMO), Rotterdam, The Netherlands, 3 Wenckebach Institute, University Medical Center Groningen, University of Groningen, Groningen, The Netherlands

* nkromme@home.nl

**Data Availability Statement:** All relevant data are within the paper and its Supporting information files.

## Abstract

Crucial to its success is that physicians enhance their competence in Lifestyle Medicine and take on their role as Health Advocates in Health Counseling and Promotion (HC&P). However, studies on patients' views of lifestyle counseling in clinical practice demonstrate that many patients neither perceived a need to adopt a healthy lifestyle nor having had any discussion with their physician about their lifestyle. This study is part of a participatory action research project focusing on identifying areas of improvement for health promotion in the practice of internists. Within this project, we interviewed 28 internists from six different subspecialties of an academic medical center in the Netherlands. This study aims to gain insight into how internists understand their role in HC&P by a qualitative analysis of their beliefs and attitudes in the interview data. Participants claimed that promoting a healthy lifestyle is important. However, they also reflected a whole system of beliefs that led to an ambivalent attitude toward their role in HC&P. We demonstrate that little belief in the success of HC&P nurtured ambivalence about the internists' role and their tasks and responsibilities. Ambivalence appeared to be reinforced by beliefs about the ability and motivation of patients, the internists' motivational skills, and the patient-doctor relationship, and by barriers such as lack of time and collaboration with General Practitioners. When participants viewed HC&P as a part of their treatment and believed patients were motivated, they were less ambivalent about their role in HC&P. Based on our data we developed a conceptual framework that may inform the development of the competences of the Health Advocate role of internists in education and practice.

**Funding:** The authors received no specific funding for this work.

**Competing interests:** The authors have declared that no competing interests exist.

## Introduction

Non-communicable chronic diseases (NCDs) are the principle cause of death and disabilities worldwide [1, 2]. In particular, the increase of multimorbidity as 'the most common chronic disease' in aging populations causes a constant rise of healthcare utilization and costs [3–6]. Healthcare delivery (including education, research and finance) has focused more than a century on infectious diseases and acute care and did not yet evolve very well into caring for patients with (multiple) chronic conditions [6, 7]. Although healthy living may prevent NCD to a large extent, a healthy lifestyle is not prevailing in the population at large [8, 9]. Therefore, the prevention of NCDs is back in the political spotlight of many western countries, including the Netherlands, where the government adopted the National Prevention Agreement [10].

In line with the renewed focus on prevention, Lifestyle Medicine (LM) emerged as a growing discipline aiming to apply and integrate evidence-based lifestyle interventions into clinical practice to lower lifestyle-related risk factors and improve the well-being of the chronically ill [11, 12]. Crucial to its success is that physicians enhance their competence in LM [13–15] and take on their role as Health Advocates (HA) in Health Counseling and Promotion (HC&P) [16].

However, studies on patients' views of lifestyle counseling in clinical practice demonstrate that many patients neither perceived a need to adopt a healthy lifestyle, nor having had any discussion with their physician about their lifestyle [17]. Studies on barriers and facilitators for HC&P in primary care indicate that physicians do discuss lifestyle with their patients but perceive barriers to performing HC&P [18–20]. The main barrier attributed to patients appeared their unwillingness and inability to change their lifestyle [19, 21]. Barriers related to physicians were lack of knowledge and counseling skills, little confidence in the effectiveness of lifestyle interventions, and their varying attitudes. Main barriers on the organization and system level were lack of time, insufficient reimbursement for HC&P and a lack of (knowledge about) referral resources [18–20, 22, 23]. Studies also reported problems with the performance of HC&P in the hospital setting. The scarcely available studies indicate that medical specialists do not prioritize lifestyle interventions [24, 25]. It also appears they do not prioritize the HA role during graduate and post-graduate education despite the introduction of competence-based learning such as the CanMEDS framework [26–28] that explicitly includes non-clinical roles like the HA role [29].

There appears to be a mismatch between a holistic health approach of patients with chronic diseases, who would benefit from LM, and current medical practice with its ongoing specialization [3, 7]. Specialization works well when it is about acute medical problems, when there is 'a clear driver of the pathology' and when a quick, effective solution is needed [30]. However, when it comes to (complex) chronic disease success turns into fragmented, expensive care that does not fit the needs of patients with multimorbidity [31, 32]. The biomedical way of clinical reasoning fails to account for the complexity of multiple interacting problems within different biomedical and psychosocial area [30, 33]. It also does not consider behavior and environmental factors sufficiently. Therefore, a shift in focus is advocated from disease and pathology toward health and behavior and to including social determinants of health [34]. In practice, this means a patient-centered, collaborative, integrated system approach [35, 36] and aligning clinical medicine, behavioral health, public health, and community services [37]. Redefining the task of medicine and the role of physicians in this way can be seen as changing its cultural heart or as a paradigm shift [7, 38] that ushers the third era of medicine with a focus on healthy aging of populations and on taking moral responsibility to solve inequities [39, 40].

Because of the needed transformational change, we started a Participatory Action Research (PAR) project with the aim to identify improvement opportunities for health promotion in the

practice of internists [41, 42]. Internists in the Netherlands are hospital-based specialists taking care of many chronically ill patients referred by their general practitioner or another medical specialist. Although internists usually do not provide primary care services, they may fulfill the HA role in the care for the chronically ill patient with complex problems [43]. As part of the PAR project, this study aims to explore the internists' perceptions of their role in Health Counseling and Promotion (HC&P).

## Methodology

### Qualitative approach and research paradigm

Lifestyle Medicine implies that physicians assume their role as Health Advocate. Because we view this as transforming the nature and culture of medicine, we use a cultural theory-driven approach to describe the professional role. According to cultural theory, a (professional) community expects that their members (e.g., physicians) integrate not only (professional) knowledge and skills required for their tasks but also the values, beliefs, norms, and attitudes that belong to these tasks to fulfill their roles and to become a competent member of the (e.g., medical) community [44]. Hence, we see a role as a coherent whole of rules of conduct, rights, obligations, and tasks that are not only based on shared knowledge and skills but also shared values, beliefs, and attitudes. We use cultural theory to investigate how values and beliefs, play a role in the internists' understandings of their role and in their attitudes toward health counseling and promotion, which reflect their mental models. See Box 1.

The theoretical framework of taking a role in HC&P is depicted in Fig 1.

We assumed that the internists' mental models influence the extent to which internists take on a role in Health Counseling and Promotion (HC&P). Because beliefs and attitudes are principal elements of mental models, we formulated the following research question: what are the internists' beliefs and attitudes about their role in HC&P, and how does this affect taking on a role in HC&P?

### Data collection methods

We conducted an interview study to explore the internists' views on their role in promoting a healthy lifestyle as part of a PAR project. The first three stages of this project in which we also

---

### Box 1. Mental models.

Mental models are cognitive representations of external reality that form associations or networks [45]. They are also called cognitive schemas or scripts [46]. They consist of what we have learned, usually think, feel, believe, value, expect and express about our roles and relationships and the skills, symbols, and behaviors that belong to a role. Based on beliefs about what we value and view as right, good, correct, proper, and appropriate, we explain reality and make sense of our experiences in different contexts [47]. Learned in specific (e.g., occupational) contexts, mental models link to goals and actions (e.g., promoting a healthy lifestyle): they implicitly and explicitly set forth goals, elicit preferences, and influence motives that guide our behavior [48]. Mental models are flexible knowledge structures. However, we call them 'cultural models' when it concerns relatively stable, taken-for-granted knowledge of the world that is widely shared by the members of a society or community and that play a role in their understandings of that world and their behavior in it [49].

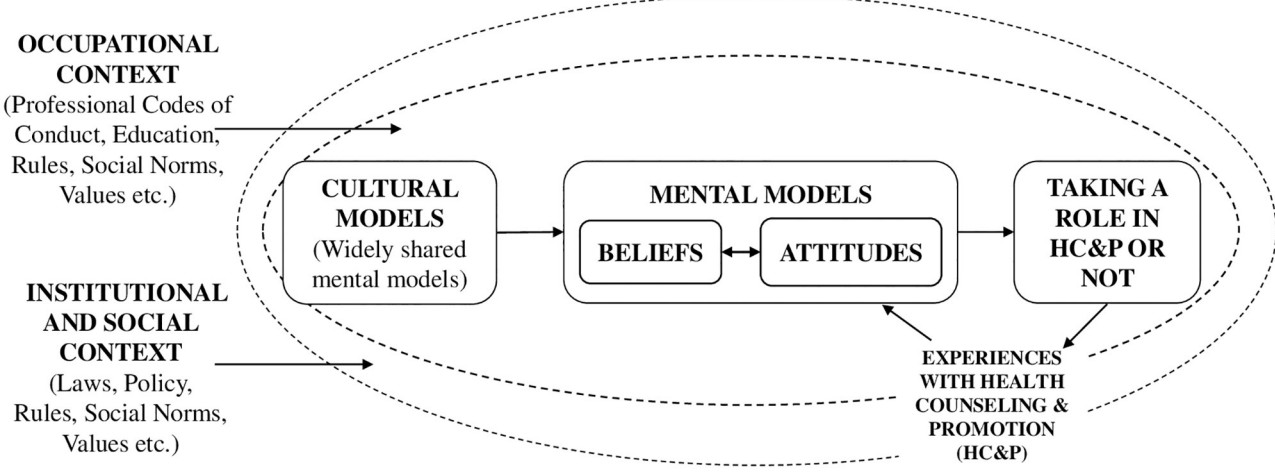

**Fig 1. Theoretical framework of taking a role in Health Counseling and Promotion (HC&P) or not.**

involved patients and other relevant care disciplines have been described elsewhere [41]. We started this project in our hospital because the Health Advocate role is part of the medical training of internists since competence-based learning -after the CanMEDS framework [16]- was introduced fourteen years ago. The HA role, however, did not fit like a glove in daily practice despite the Healthy Ageing ambition of the hospital and the Internal Medicine department [28]. Moreover, the internists appeared to perceive a number of dilemmas in lifestyle-related discussions with chronically ill patients [50]. Since there appeared to be a lack of insight in current clinical practice we started with interviewing the internists.

All interviews started with respondents filling out a statement list and were continued by asking open questions. While the statements list made it possible to gather the respondents' overt 'explicit' opinions in a short time, asking open-ended questions could explore their views that otherwise would remain hidden [51].

We based the statements and the interview questions on the following topics: issues identified in research on internists' strategies to reach productive interactions [50]; on barriers and facilitators health professionals perceive and their attitudes toward promoting a healthy lifestyle based on the theory of planned behavior [18, 19, 21, 23, 52]; on other constructs that predict behavioral change of clinicians like self-efficacy (social cognitive theory) and habit (learning theory) [53, 54].

The list of statements consisted of items such as the importance of a healthy lifestyle; the benefits of discussing a healthy lifestyle with a doctor; the tasks and responsibilities of the internist for HC&P; whose responsibility it is for patients to change their lifestyle; the patient-physician relationship, the patient's knowledge, and ability; the time of the internists, their knowledge of lifestyle interventions, their motivational skills and their role in care networks. In the interviews, we addressed similar topics by posing open-ended questions: e.g., how do you discuss lifestyle with your patients and how do you experience this as a doctor (see S1 File for the interview topic list).

The results of the statements list (see S2 File) and our first impression of the interview data pointed to a gap between the importance the internists attached to promoting a healthy lifestyle and their actual performance, indicating the internists were ambivalent about their role. Subsequently, we decided to analyze the interview data guided by our theoretical framework to gain an in-depth insight in the internists' understandings of their role in HC&P.

**Table 1. Characteristics of the responding internists.**

| Characteristics of the internists | |
| --- | --- |
| Total number included out of the whole staff * | 28 (56%) of 50 |
| Sub-specialty | |
| Acute medicine/General Internal medicine | 3 (50%) of 6 |
| Geriatrics | 3 (50%) of 6 |
| Endocrinology | 6 (67%) of 9 |
| Infectious Disease | 5 (83%) of 6 |
| Vascular Medicine/General Internal Medicine | 5 (83%) of 6 |
| Nephrology | 6 (40%) of 15 |
| Gender | |
| Men | 13 (46%) of 28 |
| Women | 15 (54%) of 28 |
| Age (and gender division) | |
| < 50 year (12 women, 8 men) | 20 (71%) of 28 |
| > 50year (3 women, 5 men) | 8 (29%) of 28 |

*NB The number includes two fellows in the last year of the internal medicine subspecialty training

## Sampling strategy

To investigate the internists' understandings of their role, we analyzed the interview data of 28 internists of six (of the seven) subspecialties of the department. We excluded the interviews within the subspecialty Allergology because we could not guarantee anonymity. The remaining sample represented sufficient variation in subspecialty, gender and age (see Table 1). Such variation is needed if one is to generate valid categories, themes, and patterns from the data collected [55].

The first author conducted all interviews by asking questions in a natural way [51]. When the participants talked about the issues they encountered in the interactions with patients, she probed further and paraphrased the responses to check that she had understood them well.

The interviews lasted 25–45 minutes and were transcribed.

## Data analysis

For the analysis of the interview transcripts and the data management, we used the Atlas.ti software version 8.4. The transcripts were carefully read, coded and analyzed following broadly the principles of grounded theory [56]. Although grounded theory offers an inductive method, our analysis also concerned deductive reasoning based on our theoretical framework and the constructs we used in the design of the statement lists and interview topic list. Next, the coding process involved an interplay between deductive and inductive coding and reasoning [55] and consisted of three steps: 1. close reading and coding; 2. categorizing codes resulting in a code list and code categories; 3. identifying relations between codes and developing code networks. For the next step we turned to a method based on a cultural theory of cognitive structures [51]. Within the established code categories we identified the quotes that reflected the respondents more or less implicit beliefs about HC&P and formulated propositions that described the essence of these beliefs. Based on the formulated propositions within these categories, schematical overviews of beliefs categories were created to demonstrate the connection between the respondents' more or less implicit beliefs and their attitudes.

Internal validation of the analysis was established by extensive discussions among the authors about the categorized propositions. This resolved any discrepancies and led to the final beliefs categories.

### Ethical approval pertaining to human subjects

The Committee for Medical Ethics of the University Medical Centre Groningen decided the study did not require ethical approval under the Dutch Medical Research Involving Human Subjects Act. Subsequently, the Scientific Commission of the Internal Medicine Department judged that the study met the quality standards of scientific research.

We asked and obtained written informed consent from all participants, and we adhere to the Standards for Reporting Qualitative Research [57].

## Results

As the main result, we found that a whole system of beliefs led to an ambivalent attitude about the internists' role in HC&P. The following paragraphs describe these beliefs within four identified categories and how these beliefs interconnect and have led to ambivalence.

To provide knowledge of the work environment of the participating internists the characteristics of the involved out-patient clinics and the available lifestyle support at these clinics are presented in Table 2. We will address in the discussion i.e., the section on the occupational context how the historically grown differences between subdisciplines of internal medicine and the availability of lifestyle support at their respective out-patient clinics, may have affected the results.

### Importance, benefits and success of HC&P and the internists' role

**Importance and benefits versus success of HC&P (Fig 2).** In the interviews, respondents found a healthy lifestyle (very) important for patients with high cardiovascular risk such as patients with chronic kidney disease, diabetes or HIV. Patients grow older, have more comorbidity, and are more often obese and/or vulnerable than several years ago. However, talking about the importance and benefits of a healthy lifestyle (in terms of better outcomes and reduction of medication), several respondents immediately expressed doubts about the success of HC&P. Despite some successful examples, most respondents expected little effect of lifestyle interventions in the long-term. They based this on the literature and their experience that long-standing unhealthy habits are hard to change and often exist in groups with a low Social Economic Position (SEP) [58]. Others emphasized the difficulty of losing weight or quitting smoking or stated that time for secondary prevention has passed and that primary prevention should have happened earlier in life.

**Having a role as an internist in HC&P (Fig 2).** Respondents who view themselves as the acting primary physician with a long-term patient-doctor relationship believe they have a role in HC&P because HC&P is part of their treatment. For instance, respondents of endocrinology or nephrology believe they have a role in HC&P for their diabetes and renal patients, and that they can influence their patients' lifestyles to a certain extent. Occasionally this holds true for respondents of other subspecialties, depending on a specific case or patient group. However, several respondents from the subspecialty infectious disease did not believe to have a role in HC&P. They believe HIV patients do not visit them for HC&P but for specific medical therapy. Several respondents from acute and vascular medicine too did not believe to have a role in HC&P in the general internal medicine outpatient clinic where they focus on diagnostic questions and usually do not have a long-term relationship with their patients. Respondents who did not believe HC&P belonged to their treatment, state that HC&P belongs to the GP

**Table 2. Characteristics of the involved outpatient clinics with regard to HC&P.**

| Sub-specialty | Outpatient clinics associated with HC&P | Available lifestyle support at these clinics |
|---|---|---|
| Acute medicine/ General Internal Medicine | General Internal Medicine*; Syncope | - |
| Geriatrics | Multimorbidity/dementia; Osteoporosis | Nurse practitioners, physical therapist |
| Endocrinology | Diabetes | Diabetes nurses, dietitians; rehabilitation program for obese diabetics |
| Infectious Disease | HIV | Nurse practitioners, social worker, dietitian |
| Nephrology | Chronic Kidney Disease (including and Hemodialysis and Renal Transplant) | Nurse practitioners, social workers, dietitians physical therapist. Recently, a lifestyle clinic and a rehabilitation program for patients with a Renal Transplant was started |
| Vascular Medicine/ General Internal Medicine | General Internal Medicine*; Vascular Disease | - |

* The General Internal Medicine out-patient clinic concerns patients referred for diagnostic questions on a wide range of internal medicine issues including Medically Unexplained Symptoms

and some also expressed the underlying belief: 'we should not overestimate our influence and role in HC&P'.

**Tasks and responsibilities in HC&P.** **Assessing risk factors and signaling lifestyle-related problems (Fig 3a)** is seen as the internists' principal task and responsibility. Respondents associated a healthy lifestyle in the first place with classical lifestyle factors such as smoking, nutrition, physical activity, overweight, and sometimes drinking alcohol. Some referred in this respect to their biomedical orientation. Most found they could easily assess smoking and exercise in contrast to food and drinking habits. They assessed stress and sleep shortly during history taking in first consultations when appraising risk factors, e.g., high blood pressure,

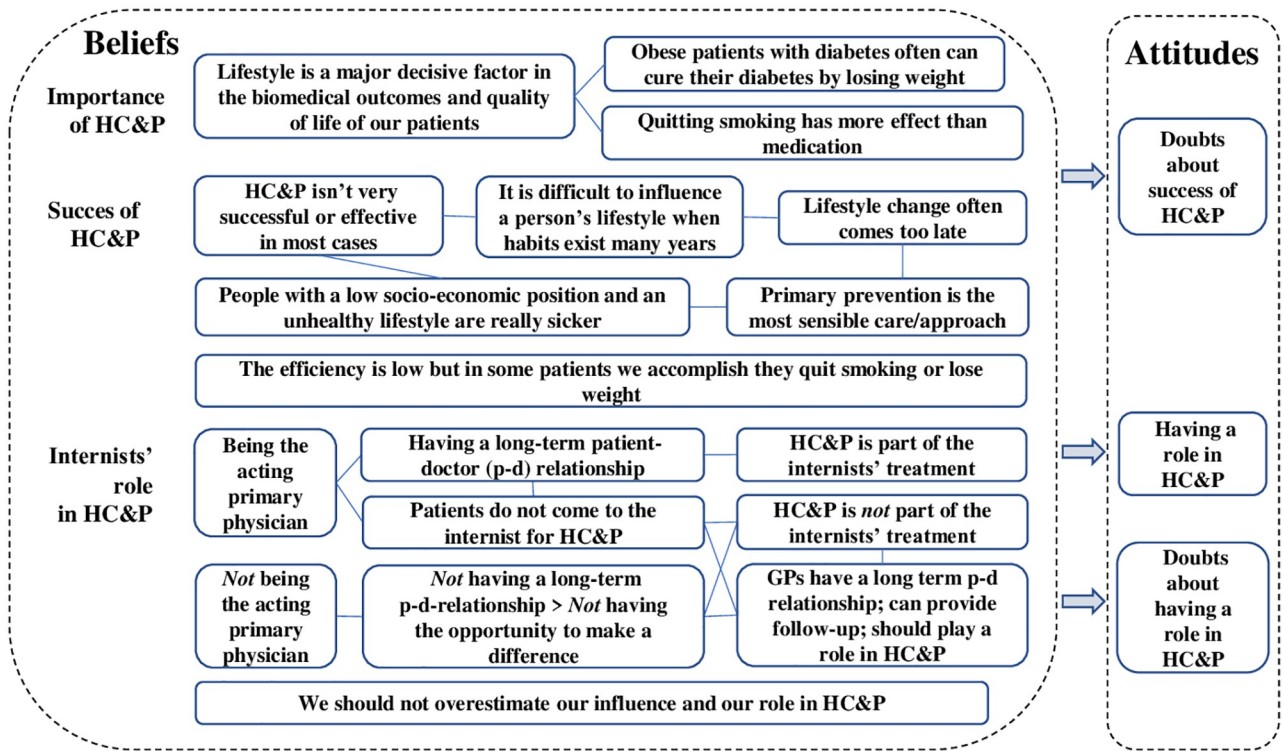

**Fig 2. Schematical overview of beliefs and attitudes about the importance and success of HC&P, and the internists' role in HC&P.**

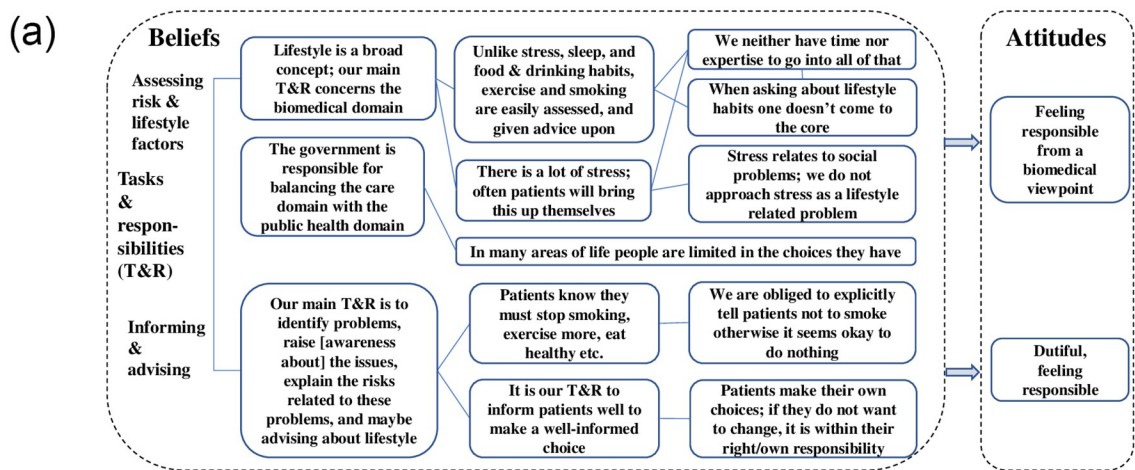

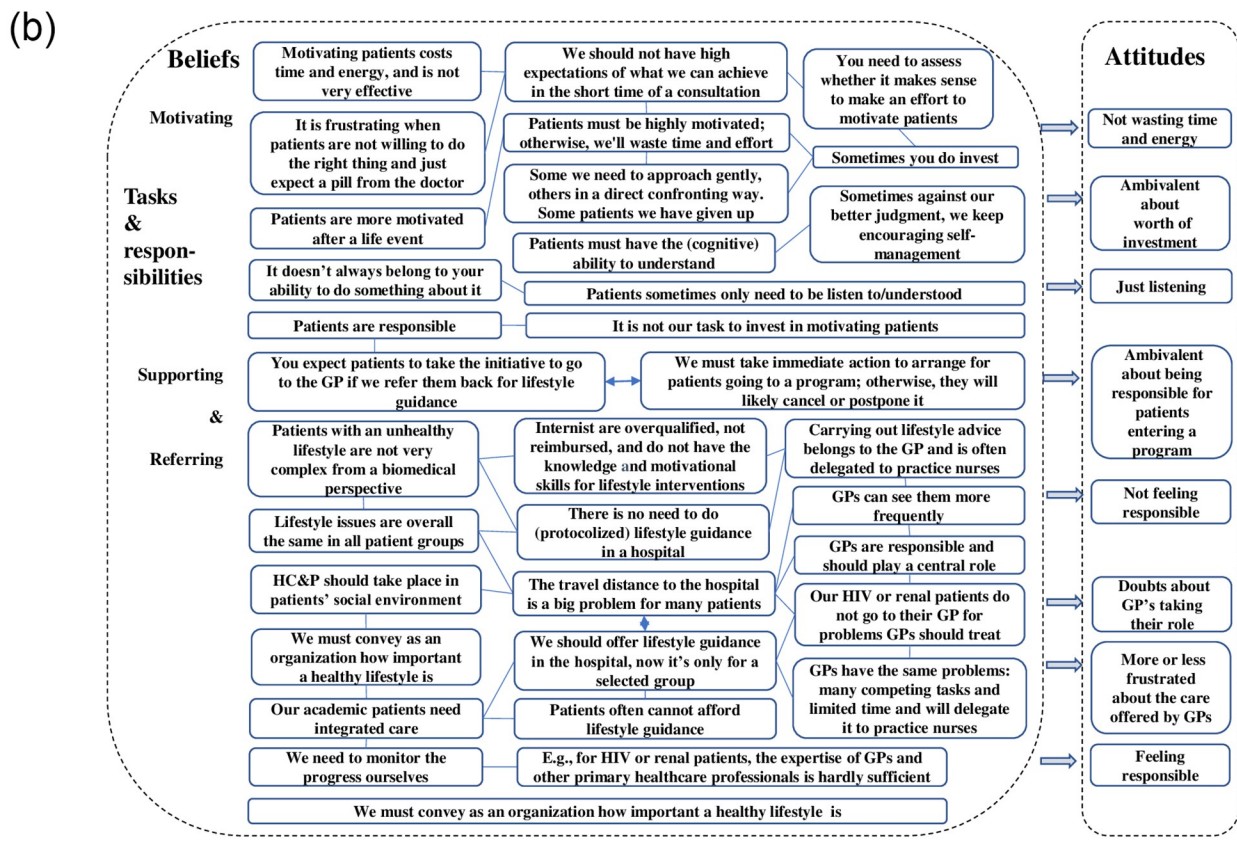

**Fig 3.** a and b. Schematical overview of beliefs and attitudes about the internists' tasks & responsibilities in HC&P.

fatigue, or obesity. Going into factors like stress or sleep in follow-up visits only occurred when the patient would bring this up. Only respondents of geriatrics said they worked according to the biopsychosocial model. They conducted a comprehensive geriatric assessment, and some said they considered sleep as a lifestyle factor. Some others mentioned they looked at the

whole person but did not view stress as a lifestyle topic but as a (psycho-) social problem that transcends lifestyle. Nevertheless, some counted stress as a lifestyle factor or pointed to socio-economic conditions that patients often cannot control. Some also mentioned that internists are responsible to point out to policymakers that they should balance the care domain with the public domain.

**Informing/explaining, and advising about lifestyle risks (Fig 3a)** respondents did view as their task and responsibility. Although they believe that patients are aware they should stop smoking, eat healthily or lose weight, several respondents believe the internist is obliged to tell patients to stop smoking. Otherwise, patients might think: it's okay because the doctor doesn't say anything about it. On the other hand, respondents believe that patients have the right to make their own choices, which means that they must ensure that patients make well-informed choices.

**Motivating patients (Fig 3b)** to lose weight or to stop smoking generally evoked ambivalent attitudes. Some respondents emphasized that patients primarily are responsible for changing their lifestyle and are free to do what they want with the advice. Some added that one should not have too high expectations of the internists' influence in the often-limited time of a visit and that one should not waste time and energy in a pointless effort. Respondents such as endocrinologists and nephrologists believe HC&P is part of their treatment. Therefore, they appeared to be more inclined to motivate patients to improve their lifestyle. Some of them also aimed on encouraging patients to self-management by giving feedback/compliments on achievements based on lab results or helping patients to set (small) achievable goals. Others said they need to confront some patients directly. Some emphasized that losing weight or stopping smoking is very difficult, and some added that patients are more motivated after a life event like a heart attack. Most respondents, however, believe patients need to be (intrinsically) motivated and to have the cognitive capacity to understand. They also said it would be wise to assess whether patients are able and ready/willing to change their lifestyle, to avoid wasting time and energy.

**Supporting and referring patients (Fig 3b)** for guidance or to a program for changing their lifestyle evoked opposite or ambivalent beliefs about the role and responsibility of the internist.

First, some respondents believe it is a patient's responsibility to go to the GP for lifestyle guidance after the internist referred them back to their GP. Others found it is the internists' responsibility to refer directly to a smoking cessation program and not lose the momentum.

Second, some respondents believe that lifestyle guidance is not their responsibility, while others believe it is an integral part of their responsibility. Arguments for not taking responsibility were that carrying out lifestyle advice is not very complicated or complex from a biomedical perspective. Hence, it belongs to the GP and can be delegated to practice nurses and should take place in the patients' community where it is cheaper. Opposite arguments were: the internist needs to monitor the progress; feeling ashamed to refer patients to a lifestyle coach when patients cannot afford it; primary care professionals lack the expertise, and specific patient groups, e.g., HIV or renal patients, do not go to their GP. Notably, several respondents expressed that GPs lack time for HC&P.

## Patient-doctor relationship; Patients' ability; Internists' motivational skills

**Patient-doctor relationship (Fig 4).** Generally, respondents believe that HC&P requires a trusting relationship and frequent follow-up visits to make a difference. Respondents also expressed what one should not do, e.g., not saying they must, not preaching, and not judging nor blaming the patient. A proper attitude should prevent patients from feeling stigmatized

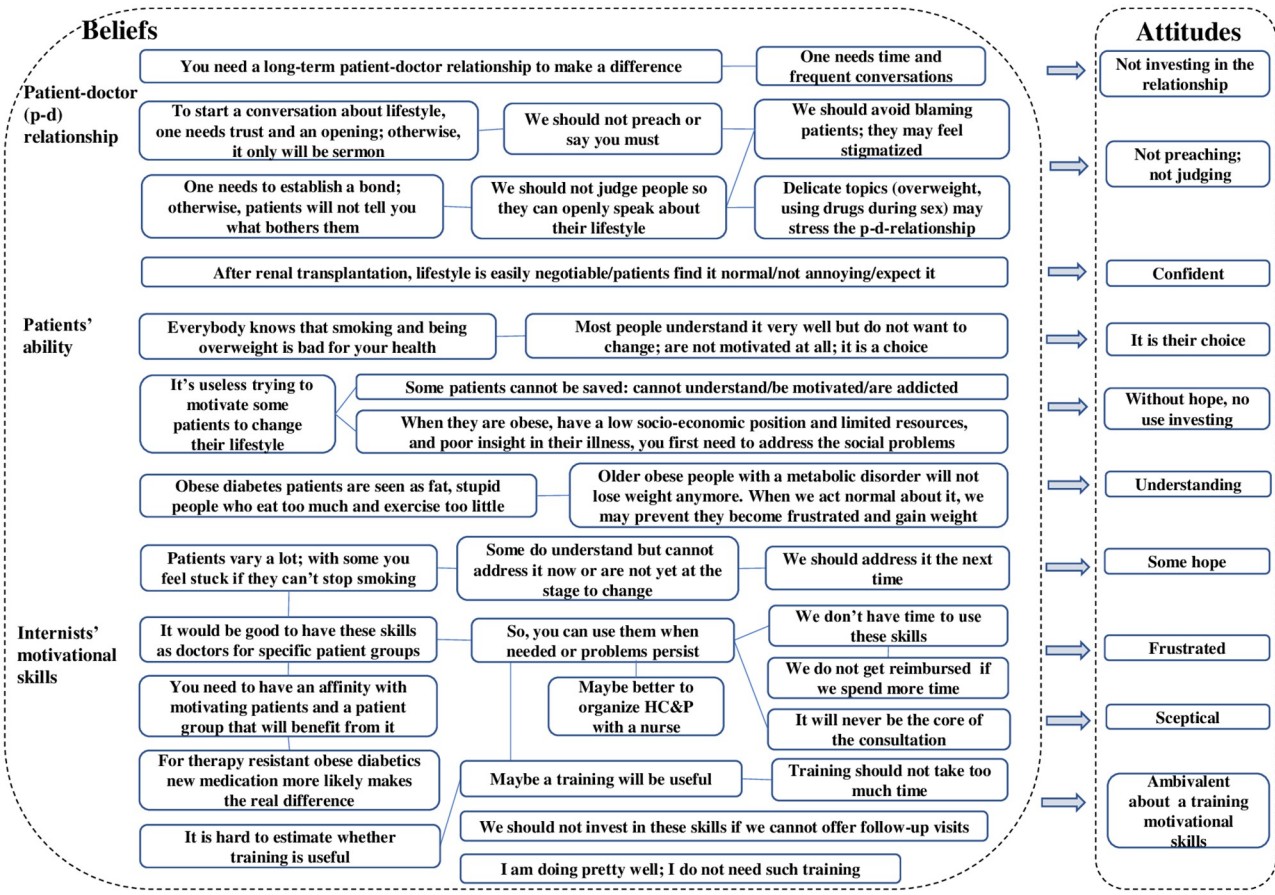

**Fig 4. Schematical overview of beliefs and attitudes about the patient-doctor relationship, the patients' ability, and the internists' motivational skills.**

and refrain from discussing smoking, being overweight, or using drugs. Some respondents, nevertheless, seemed reluctant to go into these delicate topics and to stress or disturb the relationship with their patients. Notably, some infectious diseases respondents seemed most cautious about disturbing the relation with HIV patients because they assume these patients do not want a conversation about their lifestyle and expect the doctor to focus on medical therapy. In contrast, nephrology respondents found such a conversation less disturbing or annoying. They noticed that after renal transplant patients expect such a conversation with the nephrologist.

**Patients' ability to change their lifestyle (Fig 4).** Several respondents foster the belief that patients know they should not smoke or lose their overweight but do not want to change their lifestyle, arguing it is their choice. Some added that some patients cannot be helped, based on their history, or because patients lack cognitive capacity.

Some remarked that obese diabetes patients often are stigmatized and that it would be good to understand it is difficult to lose weight in case of metabolic disease. And some stated that patients vary a lot, some are not yet in that stage or have other priorities, and that one should try it another time. As factors that influence patients' ability to change respondents mentioned:

- personal and work circumstances: difficult periods in life, heavy physical work, working in shifts, poor social-economic position, low level of education

- person-related factors: cognitive capacity, character, discipline, upbringing, ingrained habits, psychiatric problems, being sensitive to addiction

- denial or underreporting one's food intake in the case of overweight

**Internists' motivational skills (Fig 4).**   Respondents' views on their motivational skills varied, resulting in different attitudes, from frustrated or skeptical to confident and not feeling the need to invest in training.

Several believed it would be good to have these skills to use when needed, e.g., when problems persist. Others doubted whether it would make sense to invest in training. They said this also depended on the time the training would take. Some, therefore, suggested leaving HC&P to a nurse or a lifestyle coach. Again, others were not interested in training their motivational skills because they believed motivating patients is not their task or felt they did not miss communication skills to motivate for instance renal patients. Overall, however, respondents perceived time constraints as a major barrier to invest in training their motivational skills.

## Time, knowledge about interventions and collaboration in the care network

**Time (Fig 5).**   Respondents of endocrinology or nephrology believed they had not enough time in the consultation to discuss a patient's lifestyle because they needed to do many other things, such as measuring blood pressure, bodyweight, ordering lab results, charting, and making phone calls. Consequently, this meant making choices and often giving priority to the most urgent medical topics. Respondents of other subspecialties more often said they would give priority to the patients' requests for help. Some also placed lack of time in the context of a high administrative workload and, e.g., the many clicks the electronic health record requires. Some suggested more organizational support would be helpful, for instance, with recording blood pressure and lab results.

**Knowledge about interventions and collaboration in the care network (Fig 5).**   The majority of the respondents found they lacked an overview of the lifestyle interventions offered in primary care. They also had no insight into the content and quality of the interventions and what patients need to pay for them. Some, however, found they did not need to know this and that it saved time to refer patients back to the GP. Some others learned over the years where to find interventions or programs for specific patient groups. Besides missing knowledge about the interventions, respondents missed getting feedback from GPs, dietitians, and physical therapists about the progress and results of the lifestyle interventions. Some also said that collaboration with GPs was more difficult than in the early days because they no longer knew most GPs personally since the region has become so large. Investing in a care network was suggested by several respondents. However, some doubted this as well. When social problems prevail, they argued, it would be better to invest in solving these first.

**Synthesis of the results.**   We integrated the four categories of beliefs into our theoretical framework in Fig 6 (see coloured blocks). The figure shows the main beliefs within each block, how the beliefs connect, and how the whole system of beliefs results in an ambivalent attitude about the internists' role in HC&P.

We also added a block 'biomedical model and expert role' because of the participants 'biomedical view on lifestyle issues and because they understood their role and tasks as medical experts. This view led them to believe that their principal task and responsibility consisted of assessing risk factors & signalling problems and informing & advising while motivating and referring patients to change their lifestyles evoked ambivalence. Ambivalence also appeared to be caused by beliefs that HC&P is part of the treatment, lifestyle interventions are not effective in the long term, and HC&P probably will not be very successful. These beliefs influenced

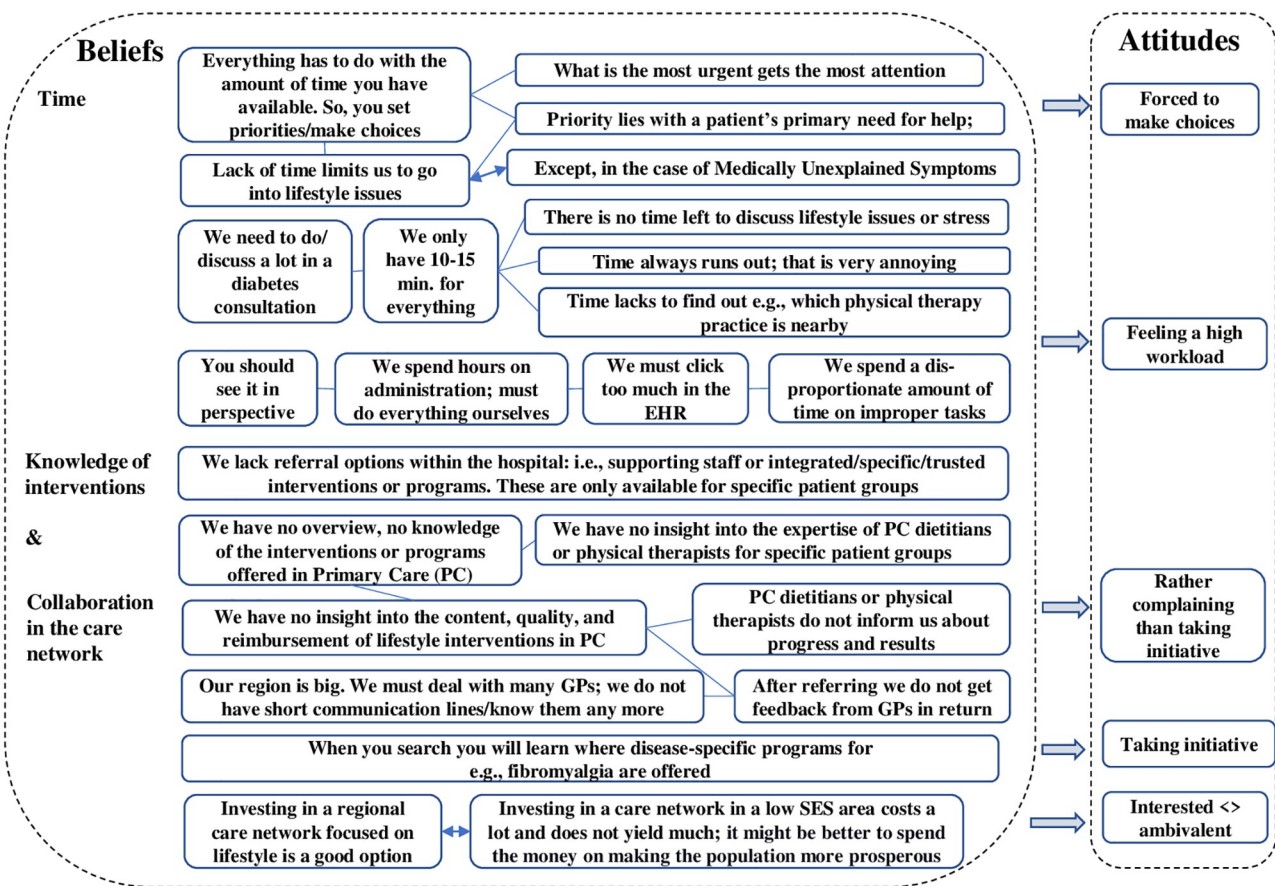

**Fig 5. Schematical overview of the beliefs and attitudes about time, the internists' knowledge of interventions and collaboration in the care network.**

beliefs about the motivating and referring tasks and connected to beliefs about the patient-doctor relationship, the patients' ability, and the internists' motivational skills. All HC&P tasks, except referring, connect to lack of time and a high administrative workload as the main barrier for not taking a role in HC&P.

## Discussion

This study aimed to gain more insight into how internists understand their role in HC&P by studying their beliefs and attitudes. As an overarching theme, we found that the respondents' beliefs about their role in HC&P evoked an overall ambivalent attitude despite the importance of health promotion and the ample evidence that motivational counseling and activating patients affect health outcomes positively [59, 60]. This broadly felt ambivalence seemed to prevent respondents from making a structural effort in HC&P and probably explains the gap between the importance attached to the role of physicians in HC&P and the internists' actual performance. Studies on attitudes of healthcare providers and on barriers and facilitators for performing or implementing HC&P identified a similar gap [18–22, 25, 61]. Most studies, however, were conducted in primary care and based on cognitive theories that explain variance in individual health behavior by ones attitude, perceived control/self-efficacy, and subjective norms [19, 23, 53, 61]. E.g., a recent cross-sectional questionnaire study on determinants

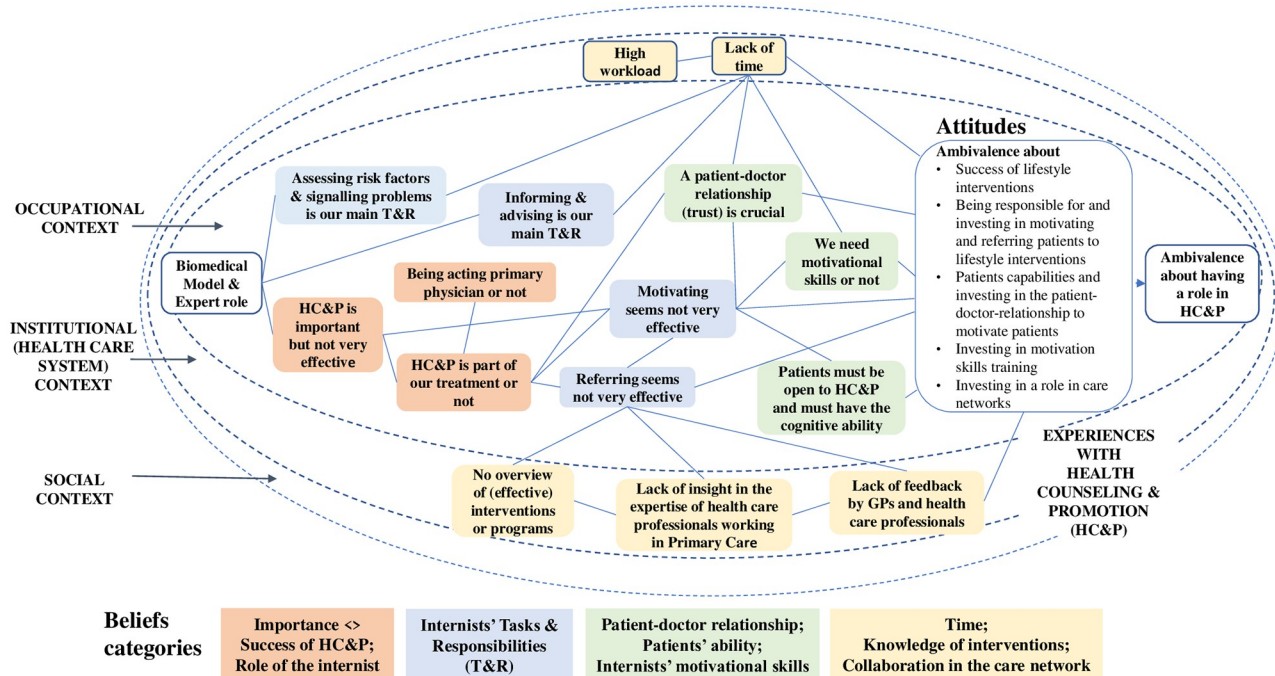

**Fig 6. Internists' beliefs and attitudes regarding their role in Health Counseling and Promotion (HC&P).**

of lifestyle counseling of Dutch GPs found that their full model of determinants (including attitudes, self-efficacy, and subjective norms) explained 47% of the variance in lifestyle counseling (LC). GPs self-efficacy (i.e., the confidence they can provide LC) was most strongly associated with LC, followed by GPs subjective norm that patients expect LC, and GPs attitude toward the use of LC [61]. The study, however, does not render insight into the GPs' more implicit beliefs and how beliefs and attitudes are connected and influenced by the context.

In contrast, the model developed in our study provides insight into how implicit beliefs interconnect and shape an ambivalent attitude about the internists' role in HC&P. Moreover, our study gives insight into how a mix of cultural, practice, and organization factors related to the occupational, institutional, and social context, can explain the respondents' ambivalence about the internists' role in HC&P.

## Occupational context

**First,** the dominance of the biomedical model that focuses on disease at the expense of patients' perceptions of their illness [62] may explain ambivalence about the HC&P role. The biomedical model can be seen as a cultural model [49] because it concerns the common knowledge that achieving excellence in biomedicine is valued and rewarding. Achieving excellence in biomedicine probably functions as the main driver to success and may explain why respondents were ambivalent about influencing a patient's behavior as a Health Advocate and engaging in care networks, which thrives on other control mechanisms, success criteria, and values.

When success depends on the patients' behavior, physicians cannot attribute success to their excellence while at the same time viewing it as frustrating when patients fail to change their lifestyle. Socialized as medical experts [63], respondents focused on diagnosing and controlling biomedical indicators, e.g., to preserve the patients' kidneys or to improve their

Hb1Ac blood levels. Although they attach value to a patients' quality of life (QoL), they may find excellence in this domain hard to achieve. Moreover, improving a patients' QoL requires looking at what matters to the patient and zooming out to the patient's health and social context. This seems opposite to their natural habit of zooming in on disease-related lifestyle factors like smoking, nutrition, physical exercise, and being overweight [33]. The same discomfort holds for their ability to use clinical reasoning to manage clinical complexity and uncertainty, added to the medical expert role, in the updated CanMEDS 2015 framework [64, 65]. Experts' natural habit is to deal with complexity by zooming in on the disease. Lifestyle-related problems, however, typically require zooming out on the patients' context and their coping with illness. Therefore, it is not surprising we found respondents did not integrate the HA-role very well with the expert role. While some argued that the complexity of chronic disease requires an integrated approach, others solved their discomfort by stating that HC&P is not complex from a medical perspective. These latter respondents expressed more ambivalence about HC&P than those who believed HC&P belongs to the task of the internist. Although some said they acted on severe social problems, supported self-management, or looked at the whole person, we also found they did not integrate social determinants of health in their lifestyle approach. Only geriatrics worked with the biopsychosocial model, and only some considered sleep and stress (and financial and social problems) to be part of patients' lifestyle determinants.

The meta-ethnography of Rubio-Valera (2014) indicates that the predominance of the biomedical model in health care hampers the implementation of prevention and health promotion in primary care [20]. Our findings point to an underlying reason: ambivalence about the doctors' role in HC&P caused by role conflict.

**Second**, experiences in their out-clinics with different patient groups may explain respondents' different degrees of ambivalence. These differences colored their mental model of what belongs to the internists' role and caused more or less ambivalence about their role and tasks in HC&P.

We found less ambivalence among respondents who felt they acted as the primary practitioners of (elderly) multimorbid diabetes or renal patients and that HC&P was thus part of their treatment. With a few exceptions, these respondents were more inclined to persist in motivating patients to change their lifestyle despite a high clinical workload, which corresponds with findings from a study on GPs counseling of multimorbid patients in primary care [66].

We, however, found more ambivalence among respondents who believe that HC&P is not part of their treatment. They viewed themselves mainly as diagnostic experts who do not have a long-term relationship with the patient and lack resources to refer patients to other health care professionals. More ambivalence we also found among infectious disease respondents. Most of them believed that HIV patients expect the doctor to focus on specific medical treatment rather than on HC&P, although HIV has become a chronic disease [67]. This finding suggests that the expectation of patients and their acceptance of lifestyle counseling influenced the respondents' attitudes toward HC&P, which is in line with other studies [61].

## Institutional context

Institutional and regional factors also may explain ambivalence about the internists' s role in HC&P, such as lack of time, lack of allocated resources, and/or lack of collaboration with GPs, dietitians, physical therapists, or lifestyle coaches working in primary care. These barriers not only may reinforce ambivalence but also may serve as justification not to take on a role in HC&P. Lack of time for HC&P appeared to be perceived as a major barrier, especially for

those respondents who perceived a high workload and had to discuss many medical issues. This finding corresponds with the general opinion that graduate and postgraduate education spend little time to develop communication skills such as active listening to patients how their illness and psycho-social circumstances affect their life (style) [68]. Lack of allocated resources frustrated respondents who believed HC&P belongs integrally to their treatment but did not have access to resources in the hospital such as dietitians or rehab programs. Lack of insight in the offered interventions in primary care and the expectation of not receiving feedback about the results, however, generally inhibited respondents to refer to primary care interventions. Partly, these findings are in line with primary care studies. It appears that GPs also perceive lack of time and a high workload as beyond their control [20, 22]. Studies also found that GPs lack motivation to refer to lifestyle interventions [22, 52] or found they do not refer to other disciplines because they perceive lack of referral options [19, 61].

## Social context

Ambivalence about the internists's role in HC&P appeared to be reinforced by the belief that patients are responsible but often not motivated to change their lifestyle. Beliefs about a person's responsibility refer to the increasing emphasis in society and healthcare that individuals have a free will and need to take care for their own health [69]. Respondents attributed low effectiveness of lifestyle interventions often to patients being reluctant or not motivated to change their lifestyle that also comes to the fore as a major barrier to treatment in primary care studies [19, 21]. Scholars, however, suggest that little success of HC&P is more easily ascribed to the patients' unwillingness than to the providers' incompetence to motivate patients [70, 71]. Probably, this is less true for physicians who take a broad approach to self-management and focus on what matters to patients in living with their condition [72].

Respondents, however, also pointed to patients' limited ability to change, which they regularly associated with a lack of (cognitive) capacity or low socio-economic position (SEP). This finding corresponds with studies on health literacy that associate health literacy with multiple health problems, a low SES, and low outcomes [73]; and with the 'capacity to think and act' [74].

In sum, respondents could be in two conflicting states of mind: On the one hand, they believe that health is a product of individual responsibility based on public values as autonomy and freedom of choice. On the other hand, they believe that health is a product of social and environmental forces, and these forces limit persons in their choices and warrant collective responsibility [75].

## Challenges

Prevention and health promotion pose challenges at the clinical and public health levels [39, 76] and require that physicians engage with the social context of their patients and interact with care professionals in a broader regional network. In response to these challenges, medical communities embraced the development of meta-competencies or non-clinical roles such as the HA-role [16, 43]. Our study, however, shows that a wide range of beliefs about the occupational, institutional, and social context affects the internists' understandings of their role in HC&P. It implies that a silver bullet approach, ensuring internists will take a HA role in HC&P, does not exist. On the contrary, the identified ambivalence and system factors point to a complex or wicked problem [77].

When internists are ambivalent about taking on a role as HA in HC&P, and ambivalence points to discomfort, and conflicting, ambiguous role understandings, it is relevant to address the causes first. This means discussing the habit of approaching chronic disease biomedically

while the complexity of chronic diseases demands a broader vision on health and taking the interaction effects between all health determinants into account by a system approach [36, 77]. It also involves discussing the responsibility of patients versus their ability to change their life-style [74]. After it is clear what role in HC&P is relevant for which patient group and internists agree about their role in HC&P, they can make decisions about task division, collaboration, care pathways, and education.

Finally, it is also about the academic work environment. Led by an efficiency perspective, such an environment may legitimate physicians to pay little time to the stories and context of patients [78].

## Strength and limitations

This is the first study that investigates academic operating internists' beliefs of their role in HC&P. Moreover, we based this study on perspectives on disease and health that challenge the current biomedical view and attribute a crucial role to physicians as Health Advocates.

The strength of this study is that we formulated research questions based on a theoretical framework. With the development of a model, we show how beliefs connect to each other and relate to attitudes, and provide insight into the internists' understandings of their role in HC&P.

It is possible that the internists who did not respond to the invitation to participate in this study were less engaged in addressing lifestyle issues than those who did. However, many internists took part in the interviews ensuring sufficient variation in subspecialty, gender and age.

It also may be a limitation we explored the internists' understanding of their role in HC&P in only one academic setting. However, our findings of their role understandings of HC&P relate to common beliefs and attitudes among physicians and may be of value to other (hospital) settings.

## Conclusion

We developed a model that shows how beliefs about the internists' role in HC&P interconnect, relate to attitudes, and are affected by the occupational, institutional, and social context. With the model, we provide new insights into the internists' understandings of their role in HC&P in academic practice. These insights may inform the development of the HA role that both patients and healthcare professionals will accept and requires a broader discussion within the public domain.

## Supporting information

**S1 File. Internists' opinions about their role in promoting a healthy lifestyle.** (PDF)

**S2 File. Interview topic list.** (PDF)

**S3 File. Minimal data set of quotes.** (PDF)

## Author Contributions

**Conceptualization:** Nicolien M. H. Kromme, Kees T. B. Ahaus, Reinold O. B. Gans, Harry B. M. van de Wiel.

**Data curation:** Nicolien M. H. Kromme.

**Formal analysis:** Nicolien M. H. Kromme, Reinold O. B. Gans, Harry B. M. van de Wiel.

**Investigation:** Nicolien M. H. Kromme.

**Methodology:** Nicolien M. H. Kromme, Harry B. M. van de Wiel.

**Project administration:** Nicolien M. H. Kromme.

**Resources:** Nicolien M. H. Kromme, Reinold O. B. Gans.

**Software:** Nicolien M. H. Kromme.

**Supervision:** Kees T. B. Ahaus, Reinold O. B. Gans, Harry B. M. van de Wiel.

**Validation:** Nicolien M. H. Kromme, Kees T. B. Ahaus, Reinold O. B. Gans, Harry B. M. van de Wiel.

**Visualization:** Nicolien M. H. Kromme, Kees T. B. Ahaus, Reinold O. B. Gans, Harry B. M. van de Wiel.

**Writing – original draft:** Nicolien M. H. Kromme, Kees T. B. Ahaus, Reinold O. B. Gans, Harry B. M. van de Wiel.

**Writing – review & editing:** Nicolien M. H. Kromme, Kees T. B. Ahaus, Reinold O. B. Gans, Harry B. M. van de Wiel.

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
