## [Decision Letter · Decision Letter 0]

4 Jul 2022

PONE-D-22-12245Internists’ ambivalent understandings of their role in health counseling and promotion:a qualitative study on the internists’ beliefs and attitudesPLOS ONE

Dear Dr. Kromme,

Thank you for submitting your manuscript to PLOS ONE. After careful consideration, we feel that it has merit but does not fully meet PLOS ONE’s publication criteria as it currently stands. Therefore, we invite you to submit a revised version of the manuscript that addresses the points raised during the review process.

I recommend that it should be revised by taking into account the changes requested by Reviewers. I want to give you a chance to revise your manuscript. The Academic Editor will only review the manuscript in the next round to speed the review process.

We look forward to receiving your revised manuscript.

Kind regards,

Baogui Xin, Ph.D.

Academic Editor

PLOS ONE

Journal Requirements:

Reviewers' comments:

Reviewer's Responses to Questions

**Comments to the Author**

1. Is the manuscript technically sound, and do the data support the conclusions?

Reviewer #1: Yes

Reviewer #2: Yes

2. Has the statistical analysis been performed appropriately and rigorously? 

Reviewer #1: N/A

Reviewer #2: Yes

3. Have the authors made all data underlying the findings in their manuscript fully available?

Reviewer #1: No

Reviewer #2: Yes

4. Is the manuscript presented in an intelligible fashion and written in standard English?

Reviewer #1: Yes

Reviewer #2: Yes

5. Review Comments to the Author

Reviewer #1: This manuscript is very relevant to the domain of health counseling and promotion; it is very well written, clearly structured, and employs a sound methodology. I also appreciate the explicit theoretical foundations of the study, which cover a wide range of determinants on various socioecological levels (individual, organizational, community), and from complementary paradigms (cognitive psychology, identity theory, attitude-behavior models). Thus, the manuscript is theoretically eclectic, while still providing the necessary focus. This shows in the excellent results section, which is very informative. The final synthesis is a real gem.

All in all, I consider this manuscript to be an asset to the literature in this field. In an attempt to further strengthen the manuscript, please consider the following minor comments:

1) In the title, I find the word 'understandings' to be a rather weak reflection of the results and the underlying theory. It is a rather cognitive and, to me, one-sided term that is not able to cover what is being put forward in the manuscript. I realize that no single word can cover all that ground, but I still request the authors to reconsider the title. For example, why not change it into 'Internists’ ambivalent beliefs and perceptions of their role in health counseling and promotion'. Or: 'Internists’ ambivalence towards their role in health counseling and promotion'.

2) From the Introduction or Methods, it does not become fully clear as to why the project focuses on internists. Could a little bit more information be provided (e.g., in the Methods) on the overarching research project? An additional reason is that reference #41 (provided in line 92) is in Dutch.

3) The section Qualitative approach and research paradigm (line #98) is rather long; perhaps this can be structured using paragraphs? For example, the explanation on mental models (line #108) can be seen as a new topic, thus starting on a new line.

4) The term 'understanding' also causes some confusion in the Methods: starting on line #121, it is stated that 'We assumed that the internists’ mental models influence their understandings of their role in Health Counseling and Promotion (HC&P) (...)'. However, based on the preceding explanation in the manuscript, it seems to make more sense to consider 'understandings' to be part of mental models, instead of their outcome. A related point is that the research question focuses on 'beliefs and attitudes', and rightly so, but what is the added value of then adding how these translate into 'understandings'? To me, these terms (attitude, beliefs, understandings) more or less represent the same phenomena, with attitude being the overarching concept (reflecting multiple cognitions as well as affective responses). My point is being confirmed by Figure 6, which represents the synthesis of the results, and which does not mention 'understandings'; this is probably because attitudes, beliefs, and other related, yet distinct, concepts are all that is needed to explain the findings. So what are 'understandings' and why do we need that concept in addition to the other (more theoretically meaningful) concepts?

5) The theoretical basis is very strong (despite my previous comment), and the choice for interviews, as well as how the interviews were performed, is well explained. However, a description of the research design seems to be omitted. Is this on purpose? Reference #54, mentioned in Data analysis, uses the grounded theory design. Is this also being employed in the manuscript, and should this be mentioned?

6) According to the authors, table 2 provides context to the study (lines #188-190), but can the authors shortly explain the meaning of that context for the study and/or results?

7) Some language or wording issues:

- line #62: that whole section is written in present time; consider writing 'patients do not perceive' instead of 'did not perceive'.

- line #154: 'interview' should be 'interviews'.

- line #165: 'program' should be 'version'.

- line #195: CKD is not explained anywhere; please consider writing it out.

- line #201: Socioeconomic Status (SES) is used, but I tend to agree with authors that argue it is better to use the term Socioeconomic Position (SEP). See, e.g., Krieger, N., Williams, D. R., & Moss, N. E. (1997). Measuring social class in US public health research: concepts, methodologies, and guidelines. Annual review of public health, 18, 341. ("(...) because "socioeconomic status" blurs distinctions between two different aspects of socioeconomic position: (a) actual resources, and (b) status, meaning prestige- or rank-related characteristics.")

Thank you for considering these comments!

Reviewer #2: This paper showed a nice topic on the beliefs and attitudes of internists about their role as Health Advocates in health counseling and promotion by qualitative method. The revision and the response to the comments of the reviewers are clear.

6. PLOS authors have the option to publish the peer review history of their article (what does this mean?). If published, this will include your full peer review and any attached files.

Reviewer #1: **Yes: **Bob C Mulder

Reviewer #2: No

---

## [Author Response · Author response to Decision Letter 0]

15 Aug 2022

Response of the author to the Journal requirements and the comments of the reviewers.

Concerning PLOS ONE Decision: PONE-D-22-15639

Internists’ ambivalent understandings of their role in health counseling and promotion: a qualitative study on the internists’ beliefs and attitudes’

A. Response to the Journal Requirements:

We addressed the following additional journal requirements as requested on behalf of our revision.

1. We followed PLOS ONE's style requirements, including those for file naming.

2. We reconsidered our indication that the data from this study are available upon request because there are no substantial legal or ethical restrictions on sharing de-identified data publicly, imposed by an ethics committee. We therefore uploaded a minimal anonymized data set as a Supporting Information file 3. We understand that our Data Availability statement will be updated based on the changed information we provided.

3. We included captions for our Supporting Information files at the end of your manuscript. `The text does not contain any in text citations referring to supporting information.

4. We checked whether our reference list is complete and correct.

NB. We added two references: One about grounded theory (nr. 56) and one about SEP (nr. 58) based on the suggestions of Reviewer # 1 (See comment 5 and 7)

B. Response to the comments of Reviewer #1

Reviewer #1: This manuscript is very relevant to the domain of health counseling and promotion; it is very well written, clearly structured, and employs a sound methodology. I also appreciate the explicit theoretical foundations of the study, which cover a wide range of determinants on various socioecological levels (individual, organizational, community), and from complementary paradigms (cognitive psychology, identity theory, attitude-behavior models). Thus, the manuscript is theoretically eclectic, while still providing the necessary focus. This shows in the excellent results section, which is very informative. The final synthesis is a real gem.

All in all, I consider this manuscript to be an asset to the literature in this field. In an attempt to further strengthen the manuscript, please consider the following minor comments:

1) In the title, I find the word 'understandings' to be a rather weak reflection of the results and the underlying theory. It is a rather cognitive and, to me, one-sided term that is not able to cover what is being put forward in the manuscript. I realize that no single word can cover all that ground, but I still request the authors to reconsider the title. For example, why not change it into 'Internists’ ambivalent beliefs and perceptions of their role in health counseling and promotion'. Or: 'Internists’ ambivalence towards their role in health counseling and promotion'.

Response

We understand the comment about the term understandings in the title in relation to comment 4, and changed the title in 'Internists’ ambivalence toward their role in health counseling and promotion, a qualitative study on the internists’ beliefs and attitudes’. See line # 2.

2) From the Introduction or Methods, it does not become fully clear as to why the project focuses on internists. Could a little bit more information be provided (e.g., in the Methods) on the overarching research project? An additional reason is that reference #41 (provided in line 92) is in Dutch.

Response

We clarified in the introduction why the project focuses on internists. See the lines # 96-100:

‘Internists in the Netherlands are hospital-based specialists taking care of many chronically ill patients referred by their general practitioner or another medical specialist. Although internists usually do not provide primary care services, they may fulfill the HA role in the care for the chronically ill patient with complex problems.’

In the section Data collection methods we provided more background information about the overarching participative action research project also because reference 42 is in Dutch. See the lines # 140-148:

‘The first three stages of this project in which we also involved patients and other relevant care disciplines have been described elsewhere (41). We started this project in our hospital because the Health Advocate role is part of the medical training of internists since competence based learning -after the CanMEDS framework (16)- was introduced fourteen years ago. The HA role, however, did not fit like a glove in daily practice despite the Healthy Ageing ambition of the hospital and the Internal Medicine department (28). Moreover, the internists appeared to perceive a number of dilemmas in lifestyle related discussions with chronically ill patients (50). Because there appeared to be a lack of insight in current clinical practice we started with interviewing the internists.’

3) The section Qualitative approach and research paradigm (line #98) is rather long; perhaps this can be structured using paragraphs? For example, the explanation on mental models (line #108) can be seen as a new topic, thus starting on a new line.

Response

We structured the section Qualitative approach and research paradigm by placing the explanation of mental models into a box. See the lines # 117-128 and the # lines 550-560 at the end of the document.

4) The term 'understanding' also causes some confusion in the Methods: starting on line #121, it is stated that 'We assumed that the internists’ mental models influence their understandings of their role in Health Counseling and Promotion (HC&P) (...)'. However, based on the preceding explanation in the manuscript, it seems to make more sense to consider 'understandings' to be part of mental models, instead of their outcome. A related point is that the research question focuses on 'beliefs and attitudes', and rightly so, but what is the added value of then adding how these translate into 'understandings'? To me, these terms (attitude, beliefs, understandings) more or less represent the same phenomena, with attitude being the overarching concept (reflecting multiple cognitions as well as affective responses). My point is being confirmed by Figure 6, which represents the synthesis of the results, and which does not mention 'understandings'; this is probably because attitudes, beliefs, and other related, yet distinct, concepts are all that is needed to explain the findings. So what are 'understandings' and why do we need that concept in addition to the other (more theoretically meaningful) concepts?

Response

We used the term understandings to describe the participants’ overall interpretation of their role in Health Counselling and Promotion based on the use of the term understanding in the literature about cultural models. See for instance p 12 of ref 48: “Cultural models of self and live organize what are, literally, vital understandings. These understandings -however, differently they may be delineated in different cultures- become again in D ’Andrade’s (ibid.) words the most general source of ‘guidance’, ‘orientation’ and ‘direction’ in the system”.

We, however, understand the reviewer comments that understandings are part of mental models and not the outcome, that attitudes and understandings seem to represent the same phenomena and that using both terms may be confusing. Based on these considerations, we not only changed the title but also changed in the Introduction the lines 99-101, and in the Methodology section the lines 113-114, 130-131, and 135.

In the lines # 99-101 we changed the description of our goal in: “to explore the internists’ perceptions of their role in Health Counseling and Promotion (HC&P) instead of “to gather insight in how they understand their role in HC&P”.

In the lines # 113-114 we changed the text to be consistent with the view that understandings are part of mental models. We changed the sentence as follows:

“We use cultural theory to investigate how values and beliefs, play a role in the internists’ understandings of their role, and in their attitudes toward health counseling and promotion, which reflect their mental models.”

In the lines # 130-131 we did not use the term “understandings of their role in HC&P” anymore as an outcome of the influence of mental models. Instead we used “take on a role in HC&P” .

In line # 135 we also adapted the research question by replacing “their understanding of this role by “taking on a role in HC&P” also because ‘taking on a role in HC&P can be seen as the outcome.

5) The theoretical basis is very strong (despite my previous comment), and the choice for interviews, as well as how the interviews were performed, is well explained. However, a description of the research design seems to be omitted. Is this on purpose? Reference #54, mentioned in Data analysis, uses the grounded theory design. Is this also being employed in the manuscript, and should this be mentioned?

Response

We, did not omit a description of the research design on purpose. In the choice of the headings in the method section we followed the Standards for Reporting Qualitative Research. The choice for interviews and their design is described in the section Data collection methods.

Given the comment of the reviewer, we, however, realize we should have been more specific in the Data analysis section about the theories /methods that informed the analysis of the interview data and adapted the data analysis section.

In the lines # 185-189 we added that the coding of the data is based on the method i.e., the general principles of grounded theory and added reference nr 56. We also added that we not only used inductive coding and reasoning as a method, but also deductive coding and reasoning based on our theoretical framework and the constructs we used to design the interview topic list.

In the lines # 190 -197 we distinguished the first three steps that we took in the coding process from the last step because this last step concerned a different method based on cultural theory of cognitive structures. We also adapted the description of this last method and step: i.e.,

“Within the established code categories, we identified the quotes that reflected the respondents more or less implicit beliefs about HC&P and formulated propositions that described the essence of these beliefs”

6) According to the authors, table 2 provides context to the study (lines #188-190), but can the authors shortly explain the meaning of that context for the study and/or results?

Response

The provided context in table 2 shows that the availability of supporting disciplines such as nurse practitioners and dietitians differs per subdiscipline. Having supporting staff means one can provide lifestyle guidance for (a part of) the patient groups in the hospital setting.

We added a few lines to explain that we will address the meaning of the historically grown differences in availability of lifestyle support services at the out-patient clinics, in the discussion section under occupational context. See the lines # 215-219:

“To provide knowledge of the work environment of the participating internists the characteristics of the involved out-patient clinics and the available lifestyle support at these clinics are presented in Table 2. We will address in the discussion, i.e., the section on the occupational context, how the historically grown differences between subdisciplines and the availability of lifestyle support at their out-patient clinics, may have affected the results”

7) Some language or wording issues:

- line #62: that whole section is written in present time; consider writing 'patients do not perceive' instead of 'did not perceive'.

- line #154: 'interview' should be 'interviews'.

- line #165: 'program' should be 'version'.

- line #195: CKD is not explained anywhere; please consider writing it out.

- line #201: Socioeconomic Status (SES) is used, but I tend to agree with authors that argue it is better to use the term Socioeconomic Position (SEP). See, e.g., Krieger, N., Williams, D. R., & Moss, N. E. (1997). Measuring social class in US public health research: concepts, methodologies, and guidelines. Annual review of public health, 18, 341. ("(...) because "socioeconomic status" blurs distinctions between two different aspects of socioeconomic position: (a) actual resources, and (b) status, meaning prestige- or rank-related characteristics.")

Response

We made textual changes in the lines # 66 (was 62) , # 174 (was 154), # 185 (was 165) and # 225 (was 195) according to the suggestions of the reviewer.

Regarding the term Socioeconomic Status (SES) line # 232 (was 201), we agree that it seems better to use Socioeconomic Position (SEP) instead, although SES is more often used. We changed SES in SEP and added the suggested reference to Krieger et al. (nr. 58)

Reviewer #2: This paper showed a nice topic on the beliefs and attitudes of internists about their role as Health Advocates in health counseling and promotion by qualitative method. The revision and the response to the comments of the reviewers are clear.

Response:

We have understood that our first revision based on the comments of reviewer 2 appears to be clear.

---

## [Editor Report · Decision Letter 1]

17 Aug 2022

Internists’ ambivalence toward their role in health counseling and promotion:a qualitative study on the internists’ beliefs and attitudes

PONE-D-22-12245R1

Dear Dr. Kromme,

We’re pleased to inform you that your manuscript has been judged scientifically suitable for publication and will be formally accepted for publication once it meets all outstanding technical requirements.

Kind regards,

Baogui Xin, Ph.D.

Academic Editor

PLOS ONE
---

## [Editor Report · Acceptance letter]

23 Aug 2022

PONE-D-22-12245R1

Internists’ ambivalence toward their role in health counseling and promotion:
a qualitative study on the internists’ beliefs and attitudes

Dear Dr. Kromme:

I'm pleased to inform you that your manuscript has been deemed suitable for publication in PLOS ONE. Congratulations! Your manuscript is now with our production department.

Kind regards,

on behalf of

Professor Baogui Xin

Academic Editor

PLOS ONE